# Evaluation of Anti-Thyroperoxidase (A-TPO) and Anti-Thyroglobulin (A-Tg) Antibodies in Women with Previous Hashimoto’s Thyroiditis during and after Pregnancy

**DOI:** 10.3390/jcm13154519

**Published:** 2024-08-02

**Authors:** Maria Angela Zaccarelli-Marino, Nuha Ahmad Dsouki, Rodrigo Pigozzi de Carvalho, Rui M. B. Maciel

**Affiliations:** 1Internal Medicine Department, University of São Paulo Medical School, São Paulo CEP 01246-000, SP, Brazil; 2Internal Medicine Department, ABC Medical School Foundation, Santo André CEP 09060-870, SP, Brazil; 3Laboratory of Molecular and Translational Endocrinology, Department of Medicine, Escola Paulista de Medicina, Federal University of São Paulo, São Paulo CEP 04022-001, SP, Brazil; nuhadile@gmail.com; 4Department of Morphology and Physiology, ABC Medical School Foundation, Santo André CEP 09060-870, SP, Brazil; r.pigozzi@hotmail.com; 5Department of Medicine, Division of Endocrinology, Escola Paulista de Medicina, Universidade Federal de São Paulo, São Paulo CEP 04022-001, SP, Brazil; rui.maciel@unifesp.br; 6Laboratory of Molecular and Translational Endocrinology, Escola Paulista de Medicina, Universidade Federal de São Paulo (EPM/UNIFESP), São Paulo CEP 04022-001, SP, Brazil

**Keywords:** human chorionic gonadotropin, pregnancy, thyroiditis, autoimmune, antithyroglobulin (A-Tg), antithyroperoxidase (A-TPO), hypothyroidism

## Abstract

**Background/Objective:** Autoimmune thyroid diseases (AITD) affect 2 to 5% of the general population. This study aimed to determine changes in activity of A-Tg and A-TPO antibodies before, during, and after pregnancy in women with previous AITD. **Methods:** This was a single-center study with a retrospective review of the medical records of 30 female patients aged 25–41 years who came to our endocrinology service in the city of Santo André, state of São Paulo, Brazil, to investigate thyroid diseases. The following data were reviewed: total triiodothyronine (totalT3), total thyroxine (totalT4), free thyroxine (FT4), thyroid-stimulating hormone (TSH), and anti-TSH receptor antibodies (anti-TSH receptor or anti-thyrotropin receptor antibodies (TRAb), anti-thyroid peroxidase (A-TPO), and anti-thyroglobulin (A-Tg)). These data were reviewed for 30 patients before and during the three trimesters of pregnancy and during the three months after pregnancy. **Results:** During gestation, we observed a progressive decrease in the blood values of A-TPO and A-Tg, which reached their lowest values in the third trimester of pregnancy, but after birth, they returned to values statistically equivalent to those before pregnancy. Analyzing the three trimesters and the post-pregnancy period, A-TPO increased 192% between the first trimester and postpartum (*p* = 0.009); it increased 627% between the second trimester and postpartum (*p* < 0.001); and it increased >1000% between the third trimester and postpartum (*p* < 0.001). There was no significant difference in the A-TPO values between the pre- and post-gestational periods (*p* = 1.00), between the first and second trimesters (*p* = 0.080), or between the second and third trimesters (*p* = 0.247). **Conclusions:** According to the results presented here, we observed changes in the activities of A-Tg and A-TPO antibodies during and after pregnancy in women with previous AITD. In women who intend to become pregnant, are pregnant, or have given birth within three months, it is essential to monitor A-TPO, A-Tg, and thyroid function as well as serum thyroid hormones and TSH to identify thyroid dysfunction in a timely manner and adjust the treatment strategy to avoid the deleterious effects of hypothyroidism on both mother and baby during and after pregnancy.

## 1. Introduction

Autoimmune thyroid diseases (AITDs) affect 2 to 5% of the general population [1]. Thyroglobulin (Tg), thyroperoxidase (TPO), and thyroid-stimulating hormone receptor (TSHR) are considered the main thyroid-specific autoantigens involved in the thyroid autoimmune response. The most prevalent AITDs are Hashimoto’s thyroiditis, which is characterized by high serum levels of antithyroglobulin (A-Tg) and antithyroperoxidase (A-TPO) antibodies and lymphocytic infiltration [2], and Graves’ disease, which involves the production of thyroid autoantibodies directed at specific antigens. Anti-TSHR, characterized by lymphocytic infiltration of variable intensities in the thyroid, is a determinant of the clinical manifestations of the disease, which can vary from hyperthyroidism to hypothyroidism [3].

Hashimoto’s thyroiditis is the most common cause of primary hypothyroidism (PH) in iodine-sufficient areas [4,5], and elevated serum A-TPO levels are present in 90% of patients, so they serve as an early clinical signal of thyroid disease [6]. Hashimoto’s thyroiditis is growing more common, and it is estimated to affect 5% of Caucasians [7]. Pathology is positive in more women than men, and its incidence increases with age, peaking in frequency between 45 and 65 years [8]. Other forms of AITD include postpartum thyroiditis, silent thyroiditis, α-interferon-induced thyroiditis, and thyroiditis that accompanies polyglandular autoimmune syndromes [3].

Studies on the genes involved in the development of AITDs have found at least six susceptibility genes whose variants were associated with AITDs: human leukocyte antigen (HLA-DR), differentiation groups (CD40), cytotoxic T lymphocyte-associated protein 4 (CTLA)-4, protein tyrosine phosphatase, nonreceptor type 22 (PTPN22), Tg, and TSHR [9].

As thyroid disorders are particularly common in women of reproductive age, AITD also have repercussions on obstetrics, and thyroid dysfunction is often found during pregnancy, sometimes as a new diagnosis [10]. According to some authors [11,12], AITD can cause hypothyroidism and increase the risk of premature birth and carcinogenesis.

The prevalence of PH is approximately 2% in iodine-sufficient areas, while overt and subclinical thyrotoxicosis occur in 0.2% and 2.5%, respectively, of pregnancies in those areas [13]. Transient gestational thyrotoxicosis can also occur during pregnancy and must be differentiated from Graves’ disease [14]. These thyroid disorders are often asymptomatic or difficult to distinguish from the features of normal pregnancy by clinical data alone. According to some authors [15,16], systematic screening of pregnant women for thyroid disorders can identify most women with subclinical thyroid disease. Subclinical hypothyroidism (SCH) is associated with an increased risk of adverse effects during pregnancy [17]. The maternal requirement for thyroid hormones increases during pregnancy [18], beginning at 4–6 weeks of gestation [19].

The increased production of maternal thyroid hormone can be regulated by high concentrations of human chorionic gonadotropin (hCG) [20], a glycoprotein hormone that has structural similarity to thyroid-stimulating hormone (TSH), an agonist of the weak TSH receptor. hCG has thyroid-stimulating activity in both experimental animals and humans [21] and was demonstrated for the first time by Burger in rats [22].

During the first trimester of pregnancy, hCG, the placental hormone initially secreted by syncytiotrophoblast cells, induces a transient increase in free thyroxine (FT4) levels, which is reflected in decreased thyroid-stimulating hormone (TSH) concentrations [23]. These pregnancy-specific changes, together with the increased demand for thyroid hormones, may expose preexisting mild thyroid dysfunction, such as in gestational thyroid disease. Even though this evolution has led to the hCG system as a backup to stimulate the thyroid gland to produce thyroid hormone [14,24], endogenous thyroid hormone variation is affected by residual thyroid tissue, by the capacity for hormone synthesis, and by thyroxine-binding globulin, anti-thyroid peroxidase (A-TPO), anti-thyroglobulin (A-Tg), iodine, and hCG. In hypothyroidism, the sensitivity of thyroid function to hCG is reduced [25,26].

Early diagnosis and correct management of thyroid dysfunction during pregnancy are essential for avoiding adverse maternal and fetal complications. Using exogenous thyroid hormones to meet maternal and fetal needs is essential for a successful pregnancy, especially during early pregnancy. According to the 2017 American Thyroid Association guidelines, a TSH value of 4.0 mU/L could be considered the upper cutoff point for diagnosing SCH during pregnancy [27]. Among all patients with hypothyroidism, a TSH concentration < 2.5 mU/L should be considered the target of control during pregnancy, and TSH levels should be evaluated six weeks after delivery [27].

According to Lee and Pearce [14], postpartum thyroiditis can occur up to 1 year after delivery and must be differentiated from other forms of thyroid dysfunction. Other authors [28,29] observed a reduction of A-Tg and A-TPO in pregnant women with chronic lymphocytic thyroiditis after the initial increase in hCG and subsequent improvement of hypothyroidism in women during pregnancy. Thyroid disease is rarely described.

This study aimed to determine changes in activity of A-Tg and A-TPO antibodies before, during, and after pregnancy in women with previous AITD.

## 2. Methods

### 2.1. Study Design

This was a single-center retrospective review of the medical records of 30 female patients at the Internal Medicine Clinic of Endocrinology in the city of Santo André, state of São Paulo (SP), Brazil, who came to the endocrinology service to investigate diseases of the thyroid (Figure 1).

### 2.2. Participants

A total of 30 adult women aged 25–41 years from the ABC region and the city of São Paulo who came to the Internal Medicine Clinic of Endocrinology in the city of Santo André, SP, Brazil, during the period of March 2018 through February 2020 were diagnosed with Hashimoto’s thyroiditis and primary hypothyroidism and were treated with thyroid hormone (levothyroxine sodium at doses of 75 μg to 125 μg, according to each patient). During the treatment of primary hypothyroidism, these women became pregnant and were followed-up throughout the pregnancy and post-gestational periods (3 months).

The research was defined starting with 30 adult women after explanation about the study.

### 2.3. Data Collection

A total of 30 adult women who were diagnosed with Hashimoto’s thyroiditis and primary hypothyroidism and who, after the treatment of hypothyroidism, became pregnant were included in this study. The laboratory data (serum analysis) reviewed from the medical records of the 30 adult women before, during, and after (3 months) pregnancy are presented in Appendix A.

The laboratory tests of the 30 adult women evaluated here were collected at the first consultation for the diagnosis of Hashimoto’s thyroiditis and hypothyroidism, and when these 30 patients became pregnant, the laboratory blood samples of thyroid hormones and TSH and A-Tg and A-TPO antibodies were requested in the last week of the first, second, and third trimesters of the gestational period and post-gestational period (3 months).

The thyroid hormones total triiodothyronine (total T3), total thyroxine (total T4), and free T4 (FT4) were measured in serum by electrochemiluminescence. Thyroid-stimulating hormone (TSH) was assessed in serum by the electrochemiluminometric method. Anti-TSH receptor antibody (anti-TSHR or TRAb) was assessed in serum by the electrochemiluminescence method. The reference values are shown in Appendix A; the A-TPO and A-Tg antibodies were detected in the serum by the chemiluminescence method, and the reference values are presented in Appendix A.

Ultrasound of the thyroid gland was performed in all patients and was requested at the first appointment using ultrasound with a high-resolution multifrequency linear transducer (7.5 MHz and 10 MHz). A homogeneous echotexture and thyroid volume (ThV) between 6.0 and 17.1 cm^3^ were considered normal and were measured by experienced diagnostic imaging professionals.

The β-hCG test was verified when the patients saw the endocrinologist after they were confirmed pregnant. It was detected in serum on the first or second day of menstrual delay and was performed by immunofluorometric method. It had a value in pregnancy > 1000 IU/L as well as the following reference values: first trimester up to 150,000 IU/L; second trimester: 3500 to 20,000 IU/L; third trimester: 5000 to 50,000 IU/L.

hCG is present throughout the gestational period, but due to the objectives of our paper, we did not evaluate the numerical values of this hormone but rather the importance of its presence throughout the gestational period. Due to the limitations of this study and due to ethical issues, we did not repeat the β-hCG test that was requested to verify pregnancy because we did not follow-up pregnant patients with this test.

Hashimoto’s thyroiditis was diagnosed in patients by A-Tg or A-TPO, who presented a heterogeneous texture and marked hypoechogenicity on ultrasound of the thyroid gland and the presence of a goiter [30]. Graves’ disease was diagnosed based on the positivity of the anti-TSH receptor antibodies (anti-TSHR or TRAb). Primary hypothyroidism and primary hyperthyroidism were diagnosed by measuring total T3, total T4, FT4, and TSH. The levels of the thyroid hormones, TSH, A-Tg, and A- TPO and anti-TSHR or TRAb were measured by Fleury Laboratories in the city of São Paulo.

Hypothyroidism and hyperthyroidism were treated by an endocrinologist with levothyroxine and thionamides such as propylthiouracil and methimazole, respectively. None of the patients received corticosteroids for treatment.

The inclusion criteria for this study were pregnant patients diagnosed with Hashimoto’s thyroiditis and primary hypothyroidism. Patients with autoimmune primary hyperthyroidism such as Graves’ disease, other thyroid diseases such as thyroid cancer, or the use of medications other than thyroid hormone, such as iodine, selenium, and zinc, were excluded from this study.

### 2.4. Ethical Approval

The objectives and methods of this study were clearly defined for all patients. This study was approved by the Research Ethics Committee of the ABC Foundation School of Medicine, SP, Brazil, and registered under number 3816100.

### 2.5. Statistical Analysis

The variables were analyzed using IBM SPSS Statistics software 26. The Kolmogorov–Smirnov test was used to test the normality of the data. The variables that exhibited a nonparametric distribution were subjected to Friedman’s ANOVA, and the significance comparisons were adjusted by Bonferroni correction for multiple tests to compare the pre-pregnancy, first trimester, second trimester, third trimester, and post-pregnancy (3 months) periods. The significance cutoff was set at *p* < 0.05.

## 3. Results

The study sample consisted of 30 women with a mean age of 32 ± 2 years. All had detectable A-TPO and A-Tg, whose serum concentrations were compared between the pre-gestational trimester, first trimester, second trimester, third trimester, and post-gestational trimester (Table 1 and Appendix A). As the pregnancy continued, the blood concentrations of A-TPO and A-Tg decreased (Figure 2), reaching their lowest values in the third trimester. After pregnancy, A-TPO and A-Tg returned to their pre-pregnancy values.

Three months after beginning pregnancy, we found a statistically significant reduction (67%) in the blood A-TPO level (*p* = 0.003). Over the course of pregnancy, the blood A-TPO concentration kept decreasing, reaching values 87% lower during the second trimester than before pregnancy (*p* < 0.001) and 95% lower in the third trimester than before pregnancy (*p* < 0.001). When comparing the A-TPO values between the analyzed trimesters and the post-gestational period, we found a 192% increase between the first trimester and post-gestational period (*p* = 0.009), 627% between second trimester and post-gestational period (*p* < 0.001), and >1000% between third trimester and post-gestational period (*p* < 0.001). There was no significant difference in the A-TPO values between the pre-and post-gestational groups (*p* = 1.00), between the first and second trimesters (*p* = 0.080), or between the second and third trimesters (*p* = 0.247) (Figure 3).

The analysis of A-Tg values between the different periods also showed reductions in these values during pregnancy. They fell 78% from the pre-gestational period to the end of the first trimester (*p* = 0.029), 92% from pre-gestational period to the second trimester (*p* < 0.001), and 96% from pre-gestational period to the third trimester (*p* < 0.001). When comparing the A-Tg values between the analyzed trimesters and the post-gestational period, we found statistical differences between the first trimester and post-gestational period (272% increase, *p* = 0.001), second trimester and post-gestational period (965% increase, *p* < 0.001), and third trimester and post-gestational period (>2000%, *p* < 0.001). There was no significant difference in the A-Tg concentration between the pre-and post-gestational periods (*p* = 1.00), between the first and second trimesters (*p* = 0.080), or between the second and third trimesters (*p* = 0.942) (Figure 3).

## 4. Discussion

Three months after the beginning of pregnancy, in the 30 pregnant women studied here, there was a statistically significant reduction of 67% in the blood level of A-TPO (*p* = 0.003) (Figure 3), and the number for A-Tg was 78% (*p* = 0.029) (Figure 3). Stagnaro-Green [31] emphasized that identifying women at risk of developing postpartum thyroiditis (PPT) may result in focused postpartum screening and that A-TPO measured in the first trimester is the ideal screening tool. PPT is a destructive autoimmune disease in women who do not have overt thyroid disease before pregnancy. Arising within one year after childbirth [32], it has a 1–22% prevalence depending on the geographical area [31,33,34].

Our results suggest a different conclusion. First, the patients presented here already had Hashimoto’s thyroiditis, with A-TPO and A-Tg levels present before pregnancy (Table 1 and Appendix A). Along with the chronological progression of pregnancy, we observed a progressive decrease in the blood values of A-TPO and A-Tg (Figure 2), which reached their lowest values in the third trimester of gestation. When measured approximately three months after pregnancy, the A-TPO and A-Tg returned to values statistically equivalent to those before pregnancy (Figure 2).

Rad and Deluxe [35] reported that postpartum thyroiditis is an autoimmune disease associated with the presence of antibodies against TPO. Approximately 30–52% of women who remain positive for A-TPO in the third trimester of pregnancy will have an 80% chance of developing postpartum thyroiditis. The screening of women at high risk of developing postpartum thyroiditis, such as through a positive A-TPO test, history of postpartum thyroiditis, and type 1 diabetes mellitus, is recommended by the clinical guidelines of the Endocrine Society. High-risk women should be evaluated for serum TSH three and six months after delivery.

Our results showed that when comparing the A-TPO values between the analyzed trimesters and the post-pregnancy period, we found a 192% increase between the first trimester and postpartum (*p* = 0.009), a 627% increase from the second trimester to postpartum (*p* < 0.001), and a> 1000% increase from the third trimester to postpartum (*p* < 0.001) (Figure 3).

Zaletel and Gaberscek [36] reported that pregnancy has a marked impact on autoimmune diseases, in which the number of regulatory T cells increases during pregnancy, and the levels of antithyroid antibodies decrease. This decrease is usually transient, and a rebound effect of antibody levels is observed six weeks after delivery. Other authors [14,37] reported that postpartum thyroiditis can occur up to 1 year after delivery and must be differentiated from other forms of thyroid dysfunction. Rad and Deluxe [35] reported that during pregnancy, the A-TPO level naturally decreases due to the immunosuppressed state of pregnancy.

According to our results, there was no significant difference in the A-TPO values between the pre-and post-gestational periods (*p* = 1.00), between the first and second trimesters (*p* = 0.080), or between the second and third trimesters (*p* = 0.247). Three months after pregnancy, there was a significant increase in A-Tg and A-TPO (Figure 2 and Figure 3), so any postpartum thyroiditis could be caused by the exacerbation of antibodies A-Tg and A-TPO. The postpartum period is characterized by a rebound of the immunotolerance induced by pregnancy [38].

hCG is released a few hours after fertilization and plays a special role in promoting the downregulation of harmful maternal immunity and regulating innate and adaptive maternal immunity, which are needed for adequate fetal growth and allow the acceptance of foreign fetal antigens [39,40]. hCG represents the first known human embryo-derived signal in maternal–fetal communication, through which the embryo influences immunological tolerance and angiogenesis at the maternal–fetal interface [41]. There is now evidence that hCG is important for maintaining pregnancy and promoting the downregulation of harmful maternal immunity [40,41].

The immunosuppressive effects of hCG were demonstrated more than thirty years ago in a murine model because hCG can stimulate lymphocytes that can depress the production of polyclonal antibodies caused by various B-cell mitogens [42] and is associated with an increase in the subset of immunosuppressive CD25+CD4+ regulatory T cells [43,44]. Later, with human lymphocytes, physiologically relevant concentrations of hCG were demonstrated to depress the induction of antibody response by a protein derivative purified from tuberculin, phytohemagglutinin, lipopolysaccharide, and pokeweed mitogen [45]. The hCG receptor, which is shared with luteinizing hormone, is expressed on T and B lymphocytes, and T lymphocytes express the hCG receptor at the mRNA level [40].

Some authors [46] reported that among women with hypothyroidism, especially those with autoimmune thyroiditis, the prevalence of PPT may be 62.1–68.0%.

According to some authors [47], PPT is defined as AITD accompanied by thyroid dysfunction in the first postpartum year. However, hormonal changes similar to those of the typical PPT pattern are reported even in women with pregestational Hashimoto’s thyroiditis treated with levothyroxine.

These observations corroborate our findings because the patients described here were diagnosed with Hashimoto’s thyroiditis and primary hypothyroidism before pregnancy (Table 1, Appendix A) and started treatment with levothyroxine sodium before pregnancy (Appendix A). When they became pregnant, they continued treatment for primary hypothyroidism throughout pregnancy and after pregnancy; they were also monitored for FT4, TSH, A-TPO, and A-Tg (Appendix A).

In this study, we did not observe in the pre-gestational, gestational, and post-gestational periods in the 30 adult women evaluated that the ages and doses of thyroid hormone used for the treatment of hypothyroidism interfered with the values of A-Tg and A-TPO (Table 1 and Appendix A; Figure 2 and Figure 3).

According to the literature [48], after treatment with levothyroxine, the A-TPO antibody titers remained within the pathological range.

Another paper [35] reported that postpartum thyroiditis can cause transient or permanent thyroid disease and suggested three clinical presentations: transient hyperthyroidism in 32% of patients, transient hypothyroidism in 43% of patients, and transient hyperthyroidism followed by hypothyroidism and euthyroidism, which is the classic form of PPT in 25% of patients. These authors concluded that among patients with primary subclinical hypothyroidism, the diagnosis of PPT was based on the occurrence of any of these three conditions within the first year after delivery, such as transient thyrotoxicosis with a TSH value < 0.27 mU/L, which is the lower limit of the reference interval in nonpregnant women; transient hypothyroidism with TSH > 4.20 mU/L, which is the upper limit of the reference interval in nonpregnant women; and only transient thyrotoxicosis or only transient hypothyroidism.

According to Amino and Arata [49], subclinical autoimmune thyroiditis exacerbates after childbirth through immunological rebound mechanisms and results in five types of thyroid dysfunction. The prevalence of postpartum thyroid dysfunction is approximately 5% in mothers in the general population. Typically, an exacerbation induces destructive thyrotoxicosis followed by transient hypothyroidism, known as postpartum thyroiditis. Permanent hypothyroidism often develops later, and patients should be followed-up once every 1–2 years. Destructive thyrotoxicosis in postpartum thyroiditis must be carefully differentiated from postpartum Graves’ disease.

Postpartum thyroiditis usually occurs 1–4 months after pregnancy, while Graves’ disease can develop 4–12 months after pregnancy. TRAb is typically positive, and thyroid blood flow is high in Graves’ disease, whereas these features are absent in postpartum thyroiditis. Patients with postpartum Graves’ disease should be treated with antithyroid drugs. In the present study, the patients presented here did not have autoimmune hyperthyroidism, as their TRAb values were negative, and they did not have transient hyperthyroidism before, during, or after pregnancy (Appendix A). Periodic testing of thyroid function is recommended after recovery from postpartum thyroiditis due to the high risk of developing permanent hypothyroidism throughout life [38].

Thyroid autoimmunity refers to the presence of A-TPO, A-Tg, TRAb, or a combination of these antibodies and is present in up to 18% of pregnant women. Postpartum thyroiditis is substantially more common in women who have thyroid antibodies during pregnancy than in those who do not.

Finally, TRAbs cross the placenta from the mother to the fetus and can cause fetal or neonatal hyperthyroidism. Therefore, women who are positive for TRAbs during pregnancy should be monitored [50].

Parker et al. [51] studied 22 women with circulating antithyroid antibodies before or during pregnancy; these patients were observed during 23 pregnancies. Titers of maternal antithyroid antibodies began to fall in the first trimester, tended to hold steady at a low level in the third trimester, and rose again during the first months after pregnancy. These antibodies were detected in the umbilical cord blood of eight neonates of nine mothers who had demonstrable circulating antithyroid antibodies at delivery but could not be detected in the umbilical cord blood of seven neonates of mothers whose antibodies titers dropped to undetectable values. Antithyroid antibodies disappeared from the blood of all the children during the first three months of life. No clinical evidence of hypothyroidism was found in the neonates or at the time of follow-up examination for up to two years after birth. These authors [51], in 1961, suggested that the increased production of hCG during pregnancy could have an effect on antibodies and that additional studies should be performed.

Thus, according to our results, throughout pregnancy, the drop in blood A-TPO concentration and its statistical significance became more pronounced, reaching values 87% lower during the second trimester than before pregnancy (*p* < 0.001) and 95% lower in the third trimester than before pregnancy (*p* < 0.001).

There was no significant difference in the A-TPO values between the pre-and post-gestational periods (*p* = 1.00), first and second trimester (*p* = 0.080), and second and third trimester (*p* = 0.247) groups (Figure 2 and Figure 3).

In this study, according to our results, we observed changes in the activities of A-Tg and A-TPO antibodies in the gestational and post-gestational periods and that the term postpartum thyroiditis could be evaluated in women with previous Hashimoto’s thyroiditis during and after pregnancy.

## 5. Conclusions

According to the results presented here, we observed changes in the activities of A-Tg and A-TPO antibodies during and after pregnancy in women with previous AITD.

In women who intend to become pregnant, women who are pregnant, and women who have given birth within three months, it is essential to monitor A-TPO and A-Tg and thyroid function as well as serum thyroid hormone and TSH levels to identify dysfunction in a timely manner and adjust the treatment strategy to avoid the deleterious effects of hypothyroidism in mother and baby during and after pregnancy.

## Figures and Tables

**Figure 1 jcm-13-04519-f001:**
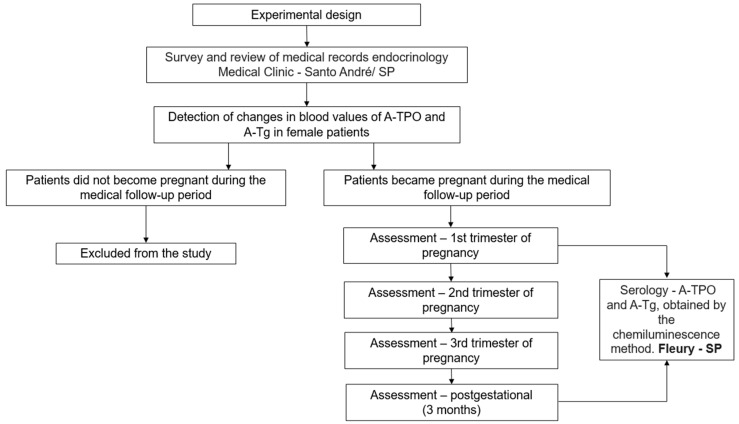
Experimental design.

**Figure 2 jcm-13-04519-f002:**
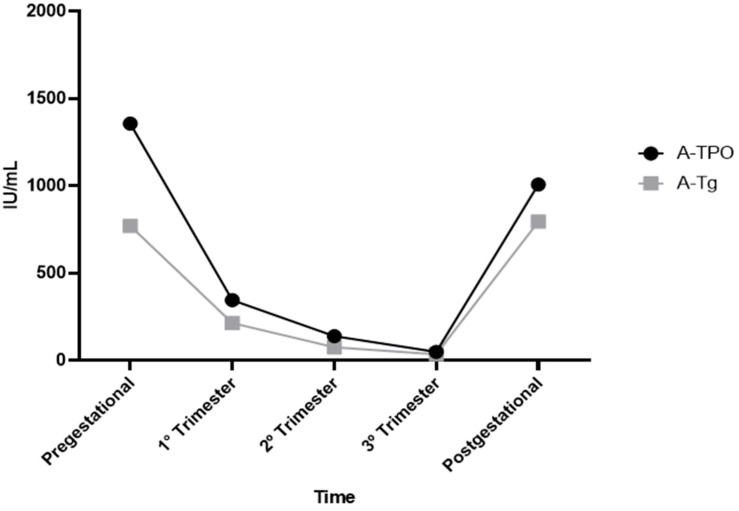
Graph showing the average behavior of the antibodies A-TPO and A-Tg over the analized periods. Reference values: A-TPO—anti thyroperoxidase antibody—negative when lower than 35 IU/mL; A-Tg—antithyroglobulin antibody—negative when lower than 40 IU/mL.

**Figure 3 jcm-13-04519-f003:**
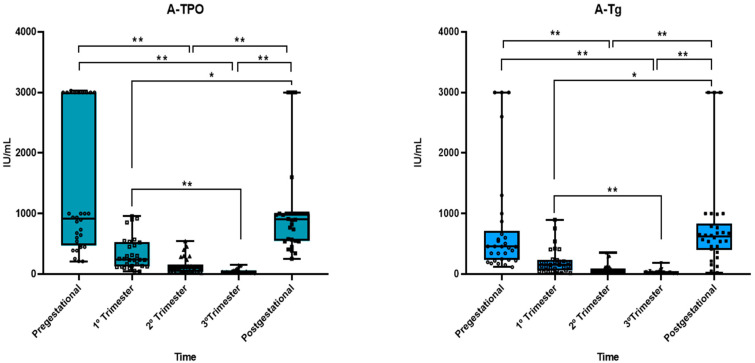
Box plot of the variations in A-TPO and A-Tg values between pregestational, first trimester, second trimester, third trimester, and postgestational. * *p* < 0.05, ** *p* < 0.001 (Friedman ANOVA adjusted by Bonferroni correction for multiple comparisons). Reference values: A-TPO—anti thyroperoxidase antibody—negative when lower than 35 IU/mL; A-Tg—antithyroglobulin antibody—negative when lower than 40 IU/mL.

**Table 1 jcm-13-04519-t001:** Comparison of A-TPO and A-Tg antibodies between pregestational, gestational, and postgestational.

Antibody	N	Period														
		Pregestational			First trimester			Second trimester			Third trimester			Postgestational		
		Median	Q1	Q3	Median	Q1	Q3	Median	Q1	Q3	Median	Q1	Q3	Median	Q1	Q3
A-TPO	30	916	471	3000	244	130	525	98	34	154	37	20	65	904	546	1000
A-Tg	30	454	233	714	156	59	235	38	23	88	26	19	38	619	401	839

Descriptive analysis of the data over the periods. Reference values: A-TPO—anti thyroperoxidase antibody—negative when lower than 35 IU/mL; A-Tg—anti thyroglobulin antibody—negative when lower than 40 IU/mL.

## Data Availability

We declare that due to ethical and privacy restrictions as this is a paper of reviewing patient records we should not put more data than those already placed here to carry out this paper.

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
