# Peer review of "Evaluation of Anti-Thyroperoxidase (A-TPO) and Anti-Thyroglobulin (A-Tg) Antibodies in Women with Previous Hashimoto’s Thyroiditis during and after Pregnancy"

_jcm, 2024, doi:10.3390/jcm13154519_

Round 1

Reviewer 1 Report

Comments and Suggestions for Authors

I am honored to have the opportunity to review the manuscript titled "Evaluation of anti-thyroperoxidase (A-TPO) and anti-thyroglobulin (A-Tg) antibodies in women with previous Hashimoto's thyroiditis during and after pregnancy".

I read the article with great interest, especially since its tone may be important in the clinical approach to patients planning a pregnancy or who are already pregnant.

Despite the above, I must draw attention to several important issues that require correction or consideration by the authors:

1) The aim of the study is to determine whether hCG interferes with aTPO and aTG antibodies, however, nowhere in the article is there an attempt to link hCG concentrations with antibody concentrations - it is clear to the reviewer that the authors used a "shortcut" related to the pregnancy, however, I believe that the purpose of the study (and therefore the conclusion) should be described more precisely, e.g. to determine changes in activity of antithyroid antibodies before, during, and after pregnancy, which is most possibly linked with hCG (or, in other words, according to the authors' wishes). Please consider this comment.

2) In Section 2.2, the authors write that the study included patients who were diagnosed with Hashimoto's thyroiditis or primary hypothyroidism. Taking into account the title and the results of additional tests presented in supplementary materials, I suggest leaving only information about patients with Hashimoto's thyroiditis.

3) The patient characteristics lack, among other things, information on the average dose of hormone used in the treatment of hypothyroidism - I suggest including this information, if available.

4) The tables mix mean (average value for numerical data assessed in parametric tests) and Q1/Q3 (measures of deviation for data assessed in non-parametric tests) - why was the median not used instead of mean (or standard deviation instead of Q1/Q3)? Please comment and verify the tabular data.

5) For each trimester and each patient, was the material for laboratory tests collected at the same timepoint during pregnancy (e.g. on a specific day or week)? If not, it may constitute a significant limitation of the manuscript, which should be included in the appropriate section of the article - it was also justified in this case to make a more detailed analysis, including a specific week of pregnancy (while, of course, leaving the current, collective, more general analysis).

6) Was it assessed how antibodies correlated with the concentrations of other laboratory parameters tested? If so, it would be worth adding it appropriately, e.g. in supplementary materials or in a separate section of the article.

7) In figures 2 and 3 - have "quarter" and "trimester" been confused?

8) Was it verified whether the age or the dose of hormone taken before and during pregnancy correlated in any way with the change in antibody concentration? These may be key elements that are worth adding to your manuscript.

9) Was it verified whether patients took supplements before or during pregnancy (including supplementation with iodine, selenium, zinc, etc.) and how could this possibly affect the results?

10) The paragraph in lines 345-348 is a bit speculative - I suggest either deleting it or covering the topic more broadly.

11) Please note numerous typos, especially a large number of missing spaces, even in the title.

The work is promising, so I think that including the above in the revised version of the manuscript will significantly improve its quality. I will be happy to read the replies to the comments and undertake a second review. Meanwhile, I congratulate the authors and wish them good luck in their further scientific endeavors.

Author Response

Reviewer 1

          I am honored to have the opportunity to review the manuscript titled "Evaluation of anti-thyroperoxidase (A-TPO) and anti-thyroglobulin (A-Tg) antibodies in women with previous Hashimoto's thyroiditis during and after pregnancy".

 Author responses:

Dear Reviewer

      We am very much appreciated the encouraging, critical, and positive comments on this manuscript. The comments have been very helpful in improving the manuscript and we have taken them fully into account in revision. Changes to the manuscript are highlighted in red.

Reviewer 1 

         I read the article with great interest, especially since its tone may be important in the clinical approach to patients planning a pregnancy or who are already pregnant.

   Author responses     

We are very grateful for your interest in considering our paper.

Reviewer 1 

        Despite the above, I must draw attention to several important issues that require correction or consideration by the authors:

Author responses     

       We appreciate your considerations and corrections.

1) The aim of the study is to determine whether hCG interferes with aTPO and aTG antibodies, however, nowhere in the article is there an attempt to link hCG concentrations with antibody concentrations - it is clear to the reviewer that the authors used a "shortcut" related to the pregnancy, however, I believe that the purpose of the study (and therefore the conclusion) should be described more precisely, e.g. to determine changes in activity of antithyroid antibodies before, during, and after pregnancy, which is most possibly linked with hCG (or, in other words, according to the authors' wishes). Please consider this comment.

Author responses                   

Lines 24,25,26,27….. Abstract 

   Objective: to determine whether hCG interferes with A-Tg and A-TPO antibodies during and after pregnancy in women with previous AITD.

       Objective: to determine changes in activity of A-Tg and A-TPO antibodies before, during and after pregnancy in women with previous AITD.

Lines  42,43,44....Conclusions

      According to the results presented here, hCG interferes with A- Tg and A-TPO antibodies during and after pregnancy in  women with previous AITD.

    According to the results presented here, we observed changes in the activities of A-Tg and A-TPO antibodies during and after pregnancy in women with previous AITD.

 Objective

Lines 120-121….This study aimed to determine whether hCG interferes with A-Tg and A-TPO antibodies during and after pregnancy in women with previous AITD.

         This study aimed to determine changes in activity of A-Tg and A-TPO antibodies before, during and after pregnancy in women with previous AITD.

2) In Section 2.2, the authors write that the study included patients who were diagnosed with Hashimoto's thyroiditis or primary hypothyroidism. Taking into account the title and the results of additional tests presented in supplementary materials, I suggest leaving only information about patients with Hashimoto's thyroiditis.

Author responses     

         Thank you very much for your observation and suggestion.

2.2. Participants

Lines 130-136…Here there was a typing error and instead of or the correct one is and, ( Line 133) as all 30 patients included in this study had Hashimoto's thyroiditis or primary  hypothyroidism, as Tables 2 and 3 of the Supplementary Material present the results of thyroid hormones and the pituitary hormone TSH.

         .

          A total of 30 adult women aged 25-41 years from the ABC region and the city of São Paulo who came to the Internal Medicine Clinic of Endocrinology in the city of Santo  André, SP, Brazil, were diagnosed with Hashimoto's thyroiditis and  primary hypothyroidism and were treated with thyroid hormone. During the treatment of primary  hypothyroidism, these women became pregnant and were followed up throughout the pregnancy and postgestational periods (3 months).

3) The patient characteristics lack, among other things, information on the average dose of hormone used in the treatment of hypothyroidism - I suggest including this information, if available.

Author responses     

         Thank you very much for your observation and suggestion. 

       Yes, it is possible to provide information on the average doses for the treatment of hypothyroidism in the 30 adult women evaluated and treated with thyroid hormone and the results do TSH are in Tables 2 of the Supplementary material before and after treatment.

2.2. Participants

Lines 131-136…

          A total of 30 adult women aged 25-41 years from the ABC region and the city of São Paulo who came to the Internal Medicine Clinic of Endocrinology in the city of Santo  André, SP, Brazil, were diagnosed with Hashimoto's thyroiditis or primary hypothyroidism and were treated with thyroid hormone. During the treatment of primary  hypothyroidism, these women became pregnant and were followed up throughout the pregnancy and postpartum periods (3 months).

        A total of 30 adult women aged 25-41 years from the ABC region and the city of São Paulo who came to the Internal Medicine Clinic of Endocrinology in the city of Santo  André, SP, Brazil, were diagnosed with Hashimoto's thyroiditis and primary hypothyroidism and were treated with thyroid hormone (levothyroxine sodium at doses of 75 μg to 125 μg, according to each patient)

          During the treatment of primary  hypothyroidism, these women became pregnant and were followed up throughout the pregnancy and postgestational periods (3 months).

4) The tables mix mean (average value for numerical data assessed in parametric tests) and Q1/Q3 (measures of deviation for data assessed in non-parametric tests) - why was the median not used instead of mean (or standard deviation instead of Q1/Q3)? Please comment and verify the tabular data.

 Author responses     

         By running the Kolmogorov‒Smirnov test, we obtained a non-parametric distribution of the data, so we expressed the data in median and Q1/Q3 intervals, by a transcription error the word mean was inserted instead of the word median, the correction was performed. Thanks for the score.

5) For each trimester and each patient, was the material for laboratory tests collected at the same timepoint during pregnancy (e.g. on a specific day or week)? If not, it may constitute a significant limitation of the manuscript, which should be included in the appropriate section of the article - it was also justified in this case to make a more detailed analysis, including a specific week of pregnancy (while, of course, leaving the current, collective, more general analysis).

 Author responses     

2.3. Data collection

Lines 137-140….The laboratory data (serum analysis) reviewed from the medical records of the 30 patients before, during, and after (3 months) gestation  are presented in Tables 2, 3, and  4 of the Supplementary Material (SM).

         The laboratory tests of the 30 adult women evaluated here were collected at the first consultation for the diagnosis of Hashimoto's thyroiditis and hypothyroidism, and when these 30 patients became pregnant, the laboratory blood samples of thyroid hormones, TSH and A-Tg and A-TPO antibodies were requested in the last week of the first, second and third trimesters of the gestational period and postgestational periods (3 months).

6) Was it assessed how antibodies correlated with the concentrations of other laboratory parameters tested? If so, it would be worth adding it appropriately, e.g. in supplementary materials or in a separate section of the article.

Author responses     

      We did not evaluate how A-Tg and A-TPO antibodies correlated with thyroid hormone and TSH concentrations.

        In the discussion we added one more reference [48] to elucidate the response to this observation.

  1. Schmidt M, Voell M, Rahlff I, Dietlein M, Kobe C, Faust M, Schicha H. Long-term follow-up of antithyroid peroxidase antibodies in patients with chronic autoimmune thyroiditis (Hashimoto’s thyroiditis) treated with levothyroxine. Thyroid(2008),18(7):755-60. doi:10.1089/thy.2008.0008

  1. Discussion

       Line 279-288

          According to some authors [47], PPT is defined as AITD accompanied by thyroid  dysfunction in the first postpartum year. However, hormonal changes similar to those of the typical PPT pattern are reported even in women with pregestational Hashimoto's  thyroiditis treated with levothyroxine.

        These observations corroborate our findings because the patients described here were  diagnosed with Hashimoto’s thyroiditis and primary hypothyroidism before pregnancy  (Table 1 and Tables 2 and 4 SM) and started treatment with levothyroxine sodium before pregnancy (Table 2 SM). When they became pregnant, they continued treatment for  primary hypothyroidism throughout pregnancy and after pregnancy; they were also  monitored for FT4, TSH, A-TPO, and A-Tg (Tables 3 and 4 SM).

       According to the literature [48], after treatment with levothyroxine, the A-TPO antibody titers remained within the pathological range.

7) In figures 2 and 3 - have "quarter" and "trimester" been confused?

 Author responses 

        Grateful for the score, the mistake was due to an error at the time of transcription of the article, however, the mistake was corrected.

8) Was it verified whether the age or the dose of hormone taken before and during pregnancy correlated in any way with the change in antibody concentration? These may be key elements that are worth adding to your manuscript.

Author responses 

          In this study, we did not observe in the pregestational, gestational and postgestational periods in the 30 adult women evaluated that the ages and doses of thyroid hormone used for the treatment of hypothyroidism interfered with the values of A-Tg and A-TPO (Table 1, Tables 2, 3, 4 SM, Figures 2 and 3).                 

      According to the literature [48], after treatment with levothyroxine, the A-TPO antibody titers remained within the pathological range.

  1. Schmidt M, Voell M, Rahlff I, Dietlein M, Kobe C, Faust M, Schicha H. Long-term follow-up of antithyroid peroxidase antibodies in patients with chronic autoimmune thyroiditis (Hashimoto’s thyroiditis) treated with levothyroxine. Thyroid(2008),18(7):755-60. doi:10.1089/thy.2008.0008

9) Was it verified whether patients took supplements before or during pregnancy (including supplementation with iodine, selenium, zinc, etc.) and how could this possibly affect the results?

        Author responses 

 Thank you very much for your observation and suggestion. 

      Yes, it was verified and we will put these observations in the exclusion factors in the methods.

    Lines 168-171 …

         The inclusion criteria for this study were pregnant patients diagnosed with Hashimoto's thyroiditis and primary hypothyroidism. Patients with autoimmune primary hyperthyroidism, such as Graves’ disease, other thyroid diseases, such as thyroid cancer, or the use of medications other than thyroid hormone, such as iodine, selenium and zinc, were excluded from this study.

10) The paragraph in lines 345-348 is a bit speculative - I suggest either deleting it or covering the topic more broadly.

         Author responses   

         Thank you very much for your observation and suggestion. 

        In this study, according to our results, we observed changes in the activities of A-Tg and A-TPO antibodies in the gestational and postgestational periods and that the term postpartum thyroiditis could be evaluated in women with previous Hashimoto's thyroiditis during and after pregnancy.

Conclusions

Lines 350-351

hCG interferes with the antibodies A-Tg and A-TPO during and after pregnancy in women with previous AITD.

       According to the results presented here, we observed changes in the activities of A-Tg and A-TPO antibodies during and after pregnancy in women with previous AITD.

11) Please note numerous typos, especially a large number of missing spaces, even in the title.

 Author responses     

         Thank you very much for your observation.

       We have corrected all the paper very carefully and also in relation to typos and the large number of missing spaces.

.

        The work is promising, so I think that including the above in the revised version of the manuscript will significantly improve its quality. I will be happy to read the replies to the comments and undertake a second review. Meanwhile, I congratulate the authors and wish them good luck in their further scientific endeavors.

Author responses     

        We are very grateful for your interest in considering our paper.

      We are looking forward to your comments and decision.

Sincerely,

Maria Angela Zaccarelli-Marino

Reviewer 2 Report

Comments and Suggestions for Authors

jcm-3127397-v1

This is an interesting study that aimed to determine whether human chorionic gonadotropin interferes with antithyroglobulin and antiperoxidase (A-TPO) antibodies during and after pregnancy in women with pre-existing autoimmune thyroid diseases. The study demonstrates that human chorionic gonadotropin interferes with the production of these antibodies during these periods. It concludes that, for women who plan to become pregnant, are pregnant, or have given birth in the last three months, it is essential to evaluate the production of antithyroglobulin and antiperoxidase antibodies, thyroid function, as well as serum levels of thyroid hormone and TSH. This will allow timely identification of dysfunctions and adjustment of treatment strategy to avoid the harmful effects of hypothyroidism on the mother and baby during and after pregnancy.

The authors present in the abstract an informative and balanced synopsis of what has been done and what has been observed.

The abstract should mention the research design with a common term.

In the introduction section, the authors explain the reasons and the scientific basis of the research, indicating the purpose of the study.

In the methods section, the authors should mention the relevant locations and dates, including follow-up and data collection periods. They should mention the type of sampling used for selecting the participants. The authors should explain how they determined the sample size. They should specify all the measures taken to address potential sources of bias.

Supplementary Table 2 shows only 29 results.

The authors need to improve the definition of the figures.

The authors should discuss the limitations of the research, taking into account possible sources of bias or imprecision. Consider both the direction and magnitude of any possible bias. Provide a cautious overall interpretation of the results considering objectives, limitations, multiplicity of analyses, results of similar studies, and other relevant empirical evidence. They should also discuss the possibility of generalizing the results.

It is necessary to verify that all citations are in brackets; for example, citation (32) is in parentheses.

Fifty percent of the bibliographic references were published more than 10 years ago.

Author Response

Reviewer 2

         This is an interesting study that aimed to determine whether human chorionic gonadotropin interferes with antithyroglobulin and antiperoxidase (A-TPO) antibodies during and after pregnancy in women with pre-existing autoimmune thyroid diseases. The study demonstrates that human chorionic gonadotropin interferes with the production of these antibodies during these periods. It concludes that, for women who plan to become pregnant, are pregnant, or have given birth in the last three months, it is essential to evaluate the production of antithyroglobulin and antiperoxidase antibodies, thyroid function, as well as serum levels of thyroid hormone and TSH. This will allow timely identification of dysfunctions and adjustment of treatment strategy to avoid the harmful effects of hypothyroidism on the mother and baby during and after pregnancy.

 Author responses:Dear Reviewer

      We am very much appreciated the encouraging, critical, and positive comments on this manuscript. The comments have been very helpful in improving the manuscript and we have taken them fully into account in revision. Changes to the manuscript are highlighted in red.

          The authors present in the abstract an informative and balanced synopsis of what has been done and what has been observed.

 Author responses:

We are very grateful for your interest in considering our paper.

The abstract should mention the research design with a common term.

Author responses

Thank you very much for your observation and suggestion

Line 24

The research design was placed in the abstract.

Study design

.

        In the introduction section, the authors explain the reasons and the scientific basis of the research, indicating the purpose of the study.

Author responses

Thank you very much for your observation.

         In the methods section, the authors should mention the relevant locations and dates, including follow-up and data collection periods. They should mention the type of sampling used for selecting the participants. The authors should explain how they determined the sample size. They should specify all the measures taken to address potential sources of bias.

Author responses

         Thank you very much for your observation and suggestion.

  1. Methods

2.2. Participants

         A total of 30 adult women aged 25-41 years from the ABC region and the city of São Paulo who came to the Internal Medicine Clinic of Endocrinology in the city of Santo André, SP, Brazil, during the period of march 2018 through february

  1.  

        The research was defined starting with 30 adult women, after the explanation about this study, were diagnosed with Hashimoto's thyroiditis and primary hypothyroidism

and were treated with thyroid hormone ( levothyroxine sodium at doses of 75 μg to 125 μg, according to each patient). During the treatment of primary hypothyroidism, these women became pregnant and were followed up throughout the pregnancy and postgestational periods (3 months).

2.3. Data collection

         A total of 30 adult women who were diagnosed with Hashimoto's thyroiditis and primary hypothyroidism and became pregnant after treatment of hypothyroidism with levothyroxine sodium at doses of 75 μg to 125 μg, according to each patient, were included in this study. 

       The laboratory data (serum analysis) reviewed from the medical records of the 30 patients before, during, and after (3 months) gestation are presented in Tables 2, 3, and 4 of the Supplementary Material (SM).

Supplementary Table 2 shows only 29 results.

Author responses

         Thank you very much for your observation.

       Supplementary Table 2 was corrected with 30 adult women.

The authors need to improve the definition of the figures.

Author responses

Thank you very much for your observation and suggestion.The definition of the figures has been improved and is placed in the text.  

          The authors should discuss the limitations of the research, taking into account possible sources of bias or imprecision. Consider both the direction and magnitude of any possible bias. Provide a cautious overall interpretation of the results considering objectives, limitations, multiplicity of analyses, results of similar studies, and other relevant empirical evidence. They should also discuss the possibility of generalizing the results.

Author responses

Objective

Lines 120-121….This study aimed to determine whether hCG interferes with A-Tg and A-TPO antibodies during and after pregnancy in women with previous AITD.

Objective

It has been improved.

This study aimed to determine changes in activity of A-Tg and A-TPO antibodies before, during and after pregnancy in women with previous AITD.

Line 137 

2.3. Data collection

Lines 153 -157

        The β-hCG test was verified when the patients saw the endocrinologist after they  were confirmed pregnant. It was detected in serum on the first or second day of menstrual  delay and was performed by immunofluorometric method. It had a value in pregnancy > 1000 IU/L and the following reference values: first trimester up to 150,000 IU/L; second  trimester: 3,500 to 20,000 IU/L; third trimester: 5,000 to 50,000 IU/L

       hCG is present throughout the gestational period, but in the objectives of our paper  we are not evaluating the numerical values of this hormone, but the importance of its presence throughout the gestational period.

        Due to the limitations of this study, due to ethical issues, we did not repeat the β-hCG test that was requested to verify pregnancy, because we did not follow pregnant patients with this test.

  1. Discussion

Line 259-266

  hCG is released a few hours after fertilization and plays a special role in promoting the downregulation of harmful maternal immunity and regulating innate and adaptive maternal immunity, which are needed for adequate fetal growth and allow the acceptance of foreign fetal antigens [39, 40]. hCG represents the first known human embryo-derived signal in maternal-fetal communication, through which the embryo influences immunological tolerance and angiogenesis at the maternal-fetal interface [41]. There is now evidence that hCG is important for maintaining pregnancy and promoting the downregulation of harmful maternal immunity [40,41].

 In the discussion we put references to the hCG.

According to the literature [48], after treatment with levothyroxine, the A-TPO antibody titers remained within the pathological range.

  1. Schmidt M, Voell M, Rahlff I, Dietlein M, Kobe C, Faust M, Schicha H. Long-term follow-up of antithyroid peroxidase antibodies in patients with chronic autoimmune thyroiditis (Hashimoto’s thyroiditis) treated with levothyroxine. Thyroid(2008) 18(7):755-60. doi:10.1089/thy.2008.0008

Lines 345-348

        We suggest that the term postpartum thyroiditis could be interpreted as pre- or postgestational Hashimoto’s thyroiditis because, according to our results, hCG could have an effect on A-TPO and A-Tg and be a factor in the differences in these antibodies observed  from the trimester of pregnancy to in the postpregnancy period.

   This sentence has been improved according to the objective of the paper.

        In this study, according to our results, we observed changes in the activities of A-Tg and A-TPO antibodies in the gestational and postgestational periods and that the term postpartum thyroiditis could be evaluated in women with previous Hashimoto's thyroiditis during and after pregnancy.

Conclusions

     Lines 350-351

hCG interferes with the antibodies A-Tg and A-TPO during and after pregnancy in women with previous AITD.

 This sentence of the conclusion has been improved.

        According to the results presented here, we observed changes in the activities of A-Tg and A-TPO antibodies during and after pregnancy in women with previous AITD.

       It is necessary to verify that all citations are in brackets; for example, citation (32) is in parentheses.

Author responses     

Line 223  …childbirth (32),

Thank you very much, it has been corrected.

  ..childbirth [32],

Fifty percent of the bibliographic references were published more than 10 years ago.

Author responses     

         In fact, many references were published more than 10 years ago and we found difficulty in more recent works published on this subject.

       We even add one more reference in this paper, also published more than 10 years ago.

  1. Schmidt M, Voell M, Rahlff I, Dietlein M, Kobe C, Faust M, Schicha H. Long-term follow-up of antithyroid peroxidase antibodies in patients with chronic autoimmune thyroiditis (Hashimoto’s thyroiditis) treated with levothyroxine. Thyroid(2008) 18(7):755-60. doi:10.1089/thy.2008.0008).

     And at the end of the paper we put an observation about postpartum thyroiditis.

           In this study, according to our results, we observed changes in the activities of A-Tg and A-TPO antibodies in the gestational and postgestational periods and that the term postpartum thyroiditis could be evaluated in women with previous Hashimoto's thyroiditis during and after pregnancy.

We are very grateful for your interest in considering our paper.

We are looking forward to your comments and decision.

Sincerely,

Maria Angela Zaccarelli-Marino

Round 2

Reviewer 1 Report

Comments and Suggestions for Authors

The authors have responded to the comments included in the previous review - the responses are satisfactory, however, it seems to me that some supplementary materials are missing. I kindly ask the authors to address this issue and, after potential verification by the Editorial Office, include the materials that are currently missing.

I suggest accepting the manuscript for publication.